

# Do human screams permit individual recognition?

Jonathan W. M. Engelberg, Jay W. Schwartz and Harold Gouzoules

Department of Psychology, Emory University, Atlanta, GA, USA

## ABSTRACT

The recognition of individuals through vocalizations is a highly adaptive ability in the social behavior of many species, including humans. However, the extent to which nonlinguistic vocalizations such as screams permit individual recognition in humans remains unclear. Using a same-different vocalizer discrimination task, we investigated participants' ability to correctly identify whether pairs of screams were produced by the same person or two different people, a critical prerequisite to individual recognition. Despite prior theory-based contentions that screams are not acoustically well-suited to conveying identity cues, listeners discriminated individuals at above-chance levels by their screams, including both acoustically modified and unmodified exemplars. We found that vocalizer gender explained some variation in participants' discrimination abilities and response times, but participant attributes (gender, experience, empathy) did not. Our findings are consistent with abundant evidence from nonhuman primates, suggesting that both human and nonhuman screams convey cues to caller identity, thus supporting the thesis of evolutionary continuity in at least some aspects of scream function across primate species.

## INTRODUCTION

For many species, the ability to recognize individuals by distinctive cues or signals is essential to the organization of social behavior (*Seyfarth & Cheney, 2015*; *Steiger & Müller, 2008*; *Tibbetts & Dale, 2007*; *Yorzinski, 2017*). Research spanning a variety of taxa, contexts, and modalities has suggested that recognition is broadly functional, facilitating behaviors directed at specific individuals (or classes thereof, such as kin) and possibly enabling opportunities to learn about third-party social dynamics (*Seyfarth & Cheney, 2015*). Recognition from vocalizations, in particular, has received attention in primates and other taxonomic groups for its potential to benefit listeners who might lack other sources of information, for example, if the vocalizer is distant or visually occluded (*Charrier, Pitcher & Harcourt, 2009*; *Seyfarth & Cheney, 2015*).

For speech or speech-derived signals, humans are adept at making identity-related judgments on the basis of vocal cues alone (*Perrachione, Del Tufo & Gabrieli, 2011*; *Remez, Fellowes & Rubin, 1997*; *Sheffert et al., 2002*; reviewed in *Mathias & Von Kriegstein, 2014*). Listeners can infer information relating to a speaker's identity even when a vocal

Corresponding author
Jonathan W. M. Engelberg,
jonathan.engelberg@emory.edu

signal has been heavily altered or reduced (*Gonzalez & Oliver, 2005*; *Remez, Fellowes & Rubin, 1997*; *Van Lancker, Kreiman & Emmorey, 1985*). However, less is known about the extent to which humans perceive and respond to identity cues in natural, nonlinguistic vocalizations such as laughter and screams. In many ways, these vocalizations seem less similar to speech than they are to the calls of nonhuman species, in terms of their acoustic structures (*Bryant & Aktipis, 2014*; *Davila-Ross, Owren & Zimmermann, 2010*; *Lingle et al., 2012*; *McCune et al., 1996*), the neural mechanisms associated with them (*Belin, 2006*; *Owren, Amoss & Rendall, 2011*), and perhaps in some of their social communicative functions as well (*McCune et al., 1996*; *Owren, Amoss & Rendall, 2011*). Research in this understudied area is therefore significant with respect to understanding the functions and evolution of the human vocal repertoire.

Thus far, research on infant cries has demonstrated that mothers and fathers can discriminate their own child's vocalizations from those of unrelated individuals (*Green & Gustafson, 1983*; *Gustafson et al., 2013*), a finding consistent with evidence from a variety of non-human taxa showing that parents recognize their own offsprings' calls (*Beecher, 1990*; *Charrier, Pitcher & Harcourt, 2009*; *Knörnschild, Feifel & Kalko, 2013*). More recent research has shown that humans can discriminate identity from laughter (although the ability to do so is impaired in spontaneous, relative to volitional, laughter; *Lavan, Scott & McGettigan, 2016*; *Lavan et al., 2018b*), and can make accurate, identity-related judgments from human "roars" (*Raine et al., 2018*). Whether human screams permit similar discrimination remains unexplored, but some comparative evidence suggests that the acoustic structures of these vocalizations might enable identity-related judgments. Similarly loud, high-pitched distress calls in species of birds (*Rohwer, Fretwell & Tuckfield, 1976*) and mammals (*Lingle, Rendall & Pellis, 2007*) elicit altruistic responses from kin, implying that they may possess features sufficient to distinguish relatedness, if not more precise individually distinctive attributes.

Many species of non-human primates produce screams primarily during agonistic conflicts (*Cheney, 1977*; *de Waal & van Hooff, 1981*). Research from our lab has documented the role of these screams in evoking aid from kin or other allies (*Gouzoules, Gouzoules & Marler, 1984*, *1986*), with significant implications for agonistic outcomes and the acquisition and maintenance of rank in the dominance hierarchy (*Cheney & Seyfarth, 1990*; *Silk, 2002*; *Gouzoules, 2005*). Importantly, these vocalizations transmit over long distances (*Gouzoules & Gouzoules, 2000*), and so are often heard by listeners lacking other situational information concerning the caller. Therefore, one might expect that screams should convey cues to kinship or identity. Evidence from numerous studies suggest that primates recognize individuals from their screams (*Bergman et al., 2003*; *Cheney & Seyfarth, 1980*; *Fugate, Gouzoules & Nygaard, 2008*; *Gouzoules, Gouzoules & Marler, 1986*; *Kojima, Izumi & Ceugniet, 2003*; *Seyfarth & Cheney, 2015*; *Slocombe et al., 2010*). Indeed, a number of playback studies investigating other cognitive abilities such as hierarchical classification (*Bergman et al., 2003*) and third-party social inference (*Slocombe et al., 2010*) have taken as a basic assumption that listeners can recognize the identity of a screamer.

Despite this behavioral evidence, some researchers have argued that screams are not acoustically well-suited for communicating identity-related cues. Most notably, Rendall
and colleagues contended that primate screams lack the individual distinctiveness of other calls, as they are too high in fundamental frequency (yielding sparser harmonic structures) and too chaotic (lacking a well-defined periodic structure) to reveal consistent, individual-specific markers of supralaryngeal filtering (*Owren & Rendall, 2003*; *Rendall, Owren & Rodman, 1998*; *Rendall, Notman & Owren, 2009*). These researchers found that rhesus macaque (*Macaca mulatta*) and human listeners alike were better able to discriminate individual macaques on the basis of their "coos"—stable, harmonically rich vocalizations—than by their screams, although, notably, humans still discriminated above chance when listening to screams (*Owren & Rendall, 2003*). A follow-up study by *Fugate, Gouzoules & Nygaard (2008)* also showed that listeners of both species were capable of discriminating rhesus macaques by their screams, but findings underscored the importance of taking into account factors that may underlie variation in response (e.g., dominance rank in macaques).

More recently, *Hansen, Nandwana & Shokouhi (2017)* acoustically analyzed human screams in an effort to develop automated systems that recognize speakers on the basis of their screams. They reported that screams exhibited substantial acoustic variability both between and within vocalizers, consistent with *Owren & Rendall's (2003)* contention that screams are, overall, unpredictable and perhaps unsuited for conveying identity. Indeed, *Hansen, Nandwana & Shokouhi (2017)* found that human listeners, when presented two successive screams, were unable to identify whether or not those screams were produced by the same vocalizer. However, this perceptual test was clearly not the primary focus of their study: only a very small number of participants ($N = 10$) completed a relatively small number of trials (10 involving two screams, whereas 10 required matching screams to speech). Thus, we suggest that it remains an open question as to whether human listeners can perceive anything about identity from screams. A more in-depth examination of human perceptual abilities in this area is needed, especially given the rich, comparative data suggesting that acoustically analogous vocalizations do permit some judgments about caller identity in other species (*Gouzoules, Gouzoules & Marler, 1986*; *Kojima, Izumi & Ceugniet, 2003*).

In the present study, participants were tested in a same-different identity discrimination task, similar to the listening tasks used in prior studies (*Fugate, Gouzoules & Nygaard, 2008*; *Hansen, Nandwana & Shokouhi, 2017*; *Owren & Rendall, 2003*), wherein they decided whether two, sequentially presented screams were produced by the same vocalizer or by two different vocalizers. The goal of this task was not to test individual recognition per se—listeners were not identifying specific individuals by their screams—but instead to test the critical prerequisite that human screams, like those of nonhuman primates, furnish perceptible cues to a vocalizer's identity. We hypothesized that if human screams convey identity cues, then listeners would perform above chance at discriminating vocalizers on the basis of screams, even if screams may lack some of the reliable, self-identifying attributes of speech or other, more periodic and/or low-pitched vocalizations.

An additional aim was to identify potential sources of variability in accuracy and response latency on the task. First, we assessed the role of both screamer and listener gender on patterns of individual recognition. Gender seems to explain some variation

in identification of emotions from non-linguistic vocalizations, with female listeners showing slightly higher accuracy than males, and vocalizer gender having mixed effects on listener accuracy (*Lausen & Schacht, 2018*). However, few studies have explored the effects of sex or gender on the auditory perception of identity in nonlinguistic vocalizations, with the exception of research on infant cries, which has focused on potential differences between mothers and fathers and has yielded mixed results (*Gustafson et al., 2013*).

In addition to gender-related factors, we investigated a possible effect of exposure to screams through the media (as assessed by a questionnaire developed in our lab); it has been suggested that experience with a vocal class may facilitate the recognition of identity from vocalizations of that kind (*Lavan et al., 2018a*). We also examined the effect of empathy (as measured by the Cambridge Behavior Scale; *Baron-Cohen & Wheelwright, 2004*), which has been shown to underlie variation in response to other distress vocalizations (*Wiesenfeld, Whitman & Malatesta, 1984*) and is particularly relevant in underpinning responses to screams (*De Waal, 2008*).

## MATERIALS AND METHODS

Testing took place over a 2-year period at Emory University's Bioacoustics Laboratory in the Department of Psychology. This research was conducted in compliance with Emory's Institutional Review Board under IRB00051516, approved July 26, 2011.

### Participants

The participant pool consisted of 104 volunteers from Emory University, including 73 females and 31 males. Ages ranged from 17 to 41 years ($M = 19.53$, SD = 2.48). The majority of participants were undergraduates, recruited via an online portal system, who received class credit for completion of the study. All participants provided their voluntary and informed written consent.

### Materials

#### Stimuli

Scream exemplars were compiled from commercial movies, scripted and unscripted television programs, newscasts, advertisements, online sources such as YouTube, and commercially available sound banks (Human Sound Effects; Partners In Rhyme, Inc., Santa Monica, CA, USA; The Nightingale Voice Box; Nightingale Music Productions, North York, Ontario, Canada). Screams comprise a salient and readily identified natural category within the human nonlinguistic vocal repertoire (*Anikin, Bååth & Persson, 2018*). Accordingly, we were able to identify screams for use in this study by ear, without utilizing formal acoustical selection criteria (which, in any event, are not fully described in the literature). In another study (*Schwartz, Engelberg & Gouzoules, in press*), participants classified as screams a large subset of the exemplars used here; all of the exemplars in that subset were classified as screams at rates of >90%. Screams were selected on the basis of sound quality (i.e., minimal noise; no overlapping sounds), and to represent a variety of emotional contexts (e.g., fear, excitement, anger, pain). Online videos were downloaded using Total Recorder version 8.0 (High Criteria, Inc., Richmond Hill, Ontario, Canada) and WinXHD

Video Converter Deluxe (Digiarty Software, Inc., Chengdu, China), while DVD media were extracted using WinXDVD Ripper Platinum (Digiarty Software, Inc., Chengdu, China). All source videos were saved, converted to the MPEG file format, and cropped at timestamps surrounding the target vocalizations.

Audio files were extracted and converted to 16-bit 22.05 kHz WAV files using Adobe Audition CC (Adobe Systems, San Jose, CA, USA) and Audacity version 2.1.2 (http://audacity.sourceforge.net). Edits were applied when necessary to delete clicks and pops (*Owren & Bachorowski, 2007*) or mitigate noise without distorting or interfering with the acoustics of the screams themselves (as determined by listening and by visual inspection of spectrograms). Additionally, in the case of some DVD sources, separate tracks containing background music were removed. Any screams that would have required more extensive editing were not used in the stimulus set. Finally, stimuli were matched for root-mean-square amplitude to achieve a uniform average presentation volume across exemplars.

Three types of stimulus pairs were constructed, each consisting of two stimuli separated by a 2,000-ms interval of silence. *Duration Modified* pairs consisted of a scream and a transformed version of itself, identical in every parameter except duration, which in the transformed version was artificially lengthened or shortened using the Stretch function in Adobe Audition. The duration of the transformed scream ranged from 50 to 150% of the original stimulus duration (*M* shortened = 71.78%; *M* lengthened = 126.91%). These trials allowed us to gauge accuracy and response latency on what we assumed would be a relatively simpler version of the task, as vocalization duration, within limits, is not typically implicated as an indexical cue of identity-related characteristics such as body size (at least compared to frequency-related parameters; *Pisanski et al., 2014*). *Same Vocalizer* pairs consisted of two different exemplars produced by the same vocalizer; these either represented different screams within the same scream bout, or different screams from entirely separate bouts and eliciting contexts. *Different Vocalizer* pairs consisted of two exemplars produced by two different vocalizers, matched for gender, relative age (e.g., child, adolescent, young adult, older adult), and apparent emotional context (including several cases in which a scream was matched with a foreign language-dubbed version produced for the exact same context). Figure 1 shows spectrograms of selected Same Vocalizer and Different Vocalizer pairs.

In all, 60 stimulus pairs were used in the experiment, comprising 12 Duration Modified trials, 24 Same Vocalizer trials, and 24 Different Vocalizer trials; this sample was comparable to that of a prior study demonstrating same-different identity discrimination for rhesus macaque screams (*Fugate, Gouzoules & Nygaard, 2008*). Note that 21 out of 90 total exemplars (and 15 out of 47 total vocalizers) were re-used in combination with different screams; the numbers of unique exemplars and vocalizers in each condition are indicated in Table 1. During data preparation for analysis, two stimulus pairs—one Same Vocalizer pair and one Different Vocalizer pair—were each found to have been duplicated (i.e., the same two exemplars matched with one another, a result of mistakenly assigning audio files two different names). We omitted duplicate trials from analyses reported here, leaving 58 stimulus pairs (12 Duration Modified, 23 Same Vocalizer,
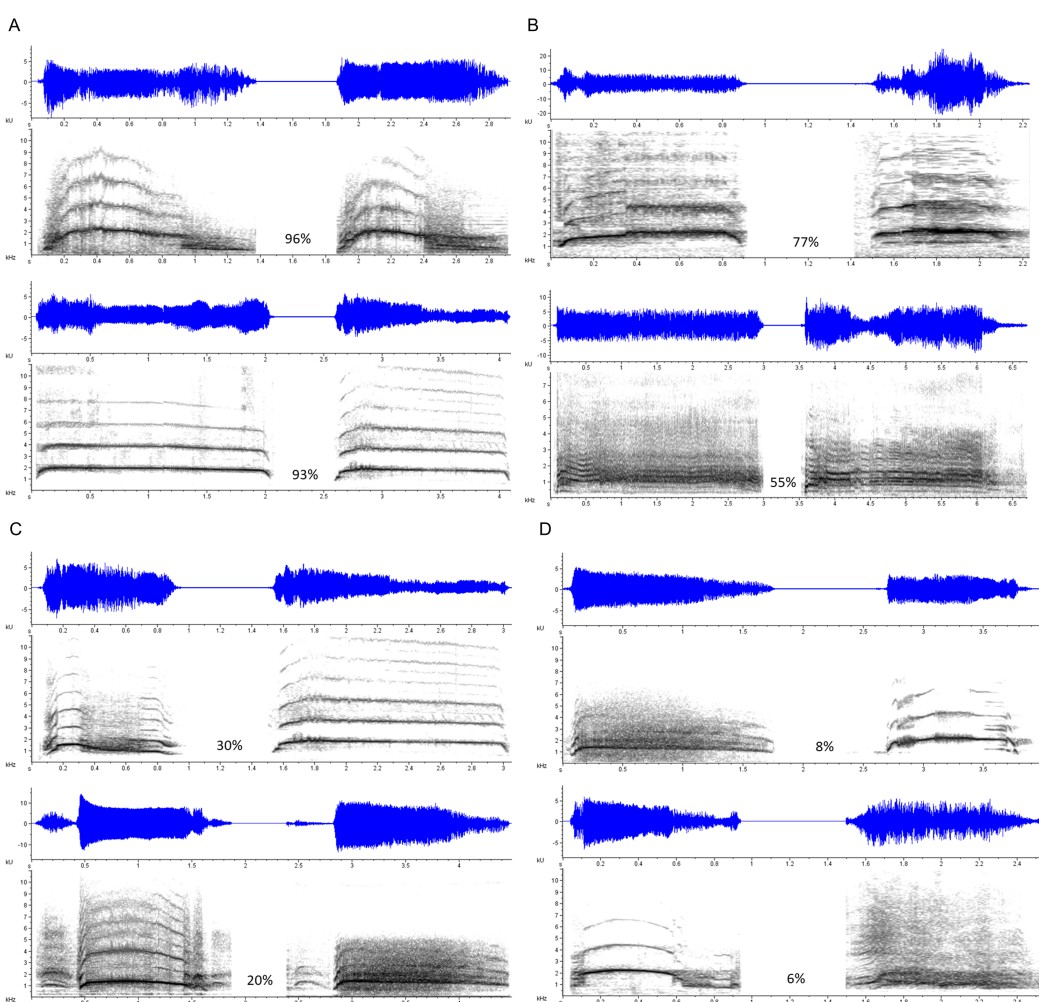

**Figure 1  Selected examples of scream pairs.** Same Scores (the proportion of participants who indicated that the pair came from the same vocalizer) noted in percentages. (A) Scream pairs by a single vocalizer that yielded high Same Scores; (B) scream pairs by different vocalizers that yielded high Same Scores; (C) scream pairs by a single vocalizer that yielded low Same Scores; (D) scream pairs by different vocalizers that yielded low Same Scores.

and 23 Different Vocalizer); however, we replicated all analyses using every possible combination of the duplicate stimulus pairs (e.g., the first Same Vocalizer pair and the second Different Vocalizer pair), and none of the study's major conclusions were affected. The few discrepancies—concerning participant factors related to variation—are noted in the Results S1.

The breakdown of stimulus pairs by vocalizer gender is shown in Table 1. The mean duration across all screams was 1,542.13 ms (ranging from 371 to 7,904 ms; SD = 1,207.60) and did not differ significantly between first and second exemplars overall ($t(56) = 0.785$, $p = 0.436$), between Duration Modified, Same Vocalizer, and Different Vocalizer trials (first exemplars: $F(2, 55) = 2.93$, $p = 0.062$; second exemplars: $F(2, 55) = 0.81$, $p = 0.448$), nor between female and male screams (first exemplars: $t(56) = 1.20$, $p = 0.235$; second exemplars: $t(56) = 0.39$, $p = 0.699$). Of the 58 stimulus pairs, 47 consisted of two acted

**Table 1 Stimulus pair information.**

| Stimulus type | Female exemplars | Male exemplars | Unique exemplars | Unique vocalizers | Number of pairs |
|---|---|---|---|---|---|
| Duration modified | 10 | 2 | 24 | 12 | 12 |
| Same vocalizer | 16 | 7 | 40 | 14 | 23 |
| Different vocalizer | 15 | 8 | 40 | 35 | 23 |
| Total | 41 | 17 | 90 | 47 | 58 |

Note:
Breakdown of vocalizer sex as well as the numbers of unique exemplars and unique vocalizers for each stimulus type. Transformed versions of screams are counted as unique exemplars in the Duration Modified pairs. The total numbers of unique vocalizers and exemplars are less than the sum of their columns because some exemplars are repeated between but not within stimulus types.

screams (taken from film, television, or a sound bank), five consisted of two naturally produced screams (taken from YouTube), and six consisted of one acted scream and one natural scream.

### Apparatus

The experiment was performed on a Sony VAIO Pentium 4 computer (model PCV-RS311, Azumino, Japan). All sounds were presented through a pair of JVC G-Series headphones (model HA-G55, JVCKENWOOD USA Corporation, Long Beach, CA, USA). Stimuli were delivered and data were collected using E-Prime 2.0 software (Psychology Software Tools, Inc., Pittsburgh, PA, USA). Participants provided input via a peripheral serial response box (model 200a, Psychology Software Tools, Inc., Pittsburgh, PA, USA); E-Prime records these responses and their latencies with millisecond-precision timing.

## Procedure

### Discrimination task

Participants were asked to judge whether each pair of screams was produced by a single vocalizer ("Same") or two different vocalizers ("Different"), indicating their responses using two labelled buttons on the serial response box. A trial began when the word "Ready" appeared in the center of the screen. After a period of 500 ms, the stimulus pair was delivered through the headphones while the screen displayed instructions reminding participants which button corresponded to which response. Participants were asked to respond as quickly and as accurately as possible, but only after the second scream in the pair had finished playing. When an input was received, an inter-trial interval of 1,000 ms proceeded with no visual or auditory presentation. All participants heard the same 60 stimulus pairs, but trials were presented in a fully randomized order. An experimenter remained in another part of the testing room to make sure instructions were followed, and to offer clarification on the task as necessary, but otherwise no feedback was provided at any time.

### Questionnaires

Following the discrimination task, personal and demographic information was collected with the use of two self-report questionnaires. The first was a 10-item survey developed specifically for studies in our lab. Seven items were designed to estimate the participant's

prior exposure to screams in the media, based on his or her agreement with statements regarding the knowledge and consumption of video games, television, and film genres likely to contain screams (e.g., "I watch TV shows where screaming often occurs"). All answers were entered on a five-point scale, where higher ratings indicated greater knowledge or familiarity. The remaining three items gauged the participant's confidence in making judgments during the experiment and in reading emotions in general (e.g., "I consider myself good at reading emotions in people"). Additionally, information regarding gender, age, first spoken language, and handedness was collected (Materials S1).

Empathy was measured using the Cambridge Behaviour Scale (*Baron-Cohen & Wheelwright, 2004*), a 40-item questionnaire that assesses cognitive as well as emotional aspects of empathic ability. The resultant Empathy Quotient (EQ) ranges from 0 to 80, with higher EQs corresponding to greater levels of empathy.

## Analysis

### Overall accuracy and response latencies

Primary outcome variables derived from the discrimination task included response accuracy (i.e., the proportion of accurate responses) and the corresponding discriminability index ($d'$) for each participant. The $d'$ statistic is used in signal detection theory to estimate a participant's sensitivity to the difference between two different stimulus groups; it is an advantageous measure over percent accuracy because it separates out the effects of a participant's response biases, which is especially important when stimulus groups are not completely balanced (*Macmillan & Creelman, 2004*). Each participant's $d'$ score was calculated as the difference between the Z-transformed *Hit Rate* and the Z-transformed *False Alarm Rate*, where a Hit equated to correctly responding "Same" when the two screams were produced by the same vocalizer (i.e., Duration Modified and Same Vocalizer trials) and a False Alarm entailed incorrectly responding "Same" when two different vocalizers had produced the screams (i.e., Different Vocalizer trials).

Mean response latencies were also calculated for each participant, using only the trials for which the participant responded correctly (*Bruyer & Brysbaert, 2011*). Responses provided before the second exemplar had finished playing were not included in the analyses reported here, as they were likely to reflect accidental or anticipatory inputs rather than decisions informed by the full comparison of both exemplars. This resulted in the exclusion of 489 trials out of 6,032 trials across all participants. With two exceptions, noted in Results S1, findings did not differ for these analyses and a separate set in which the only exclusions were responses entered before the second exemplar had started playing (precluding any comparison between the exemplars; 26 exclusions).

### Participant characteristics

A principal components analysis (PCA) with varimax rotation was conducted on the Spearman's correlation matrix obtained from participants' responses to the questionnaire items. PCA was employed to reduce the number of variables—thereby lowering the number of tests in subsequent regressions—because some of the items were correlated with

**Table 2 Results of PCA on questionnaire responses.**

**(A) Rotated components matrix from PCA on questionnaire responses**

| Questionnaire item | F1 | F2 | F3 | F4 |
|---|---|---|---|---|
| Watch movies | 0.105 | 0.05 | −0.154 | **0.856** |
| Enjoy scary movies | **0.876** | 0.052 | 0.137 | −0.023 |
| Watch scary movies | **0.839** | 0.134 | 0.135 | 0.118 |
| Watch TV with screams | **0.728** | −0.049 | 0.007 | 0.143 |
| Know about movie actors | 0.103 | −0.059 | 0.208 | **0.838** |
| Know about video games | 0.047 | **0.953** | 0.064 | 0.003 |
| Play action video games | 0.059 | **0.943** | 0.14 | −0.01 |
| Confidence about judgments | 0.241 | 0.238 | **0.732** | −0.058 |
| Difficulty making choices | −0.035 | 0.197 | **0.725** | −0.003 |
| Reading emotions | 0.089 | −0.132 | **0.613** | 0.079 |

**(B) Summary of PCA results on questionnaire responses**

| Factor | Eigenvalue | Variance explained (%) | Cumulative variance explained (%) |
|---|---|---|---|
| F1. Media | 2.708 | 27.08 | 27.08 |
| F2. Video games | 1.884 | 18.84 | 45.92 |
| F3. Confidence | 1.311 | 13.11 | 59.03 |
| F4. Movies | 1.176 | 11.76 | 70.78 |

**Note:**
Factor loadings > 0.4 are bolded.
(A) Rotated components matrix with factor loadings. (B) Summary of PCA results including all PCs with eigenvalues > 1.

each other (likely due to probing the same underlying construct; *O'Rourke & Hatcher,* *2013*). The analysis yielded four factors with eigenvalues greater than 1, cumulatively explaining 70.78% of the variance in questionnaire responses (see Table 2). Based on the factor loadings for each questionnaire item, our interpretation of each factor is as follows: F1 (Media) captures exposure to media likely to contain screams, that is, scary movies and television; F2 (Games) captures the tendency to play video games; F3 (Confidence) captures confidence in task performance and in reading emotions generally; and F4 (Movies) captures exposure to movies generally. To assess the effects of these factors on accuracy and response latency, participants' factor scores were saved and used as predictor variables, along with EQ, in multiple linear regressions with $d'$ scores and reaction times as the outcome variables.

Statistical analyses were conducted using SPSS Statistics version 24 (IBM Corp., Armonk, NY, USA). All statistical tests were two-tailed (where applicable) with a family-wise $\alpha = 0.05$.

# RESULTS

## Discrimination accuracies and response latencies: same or two different vocalizers

The mean response accuracy across all participants and stimuli was 0.77 (SE = 0.01), corresponding to a mean $d'$ score = 1.63 (SE = 0.06). In signal detection theory, $d' = 0$ indicates that participants failed to show discrimination on the tested perceptual

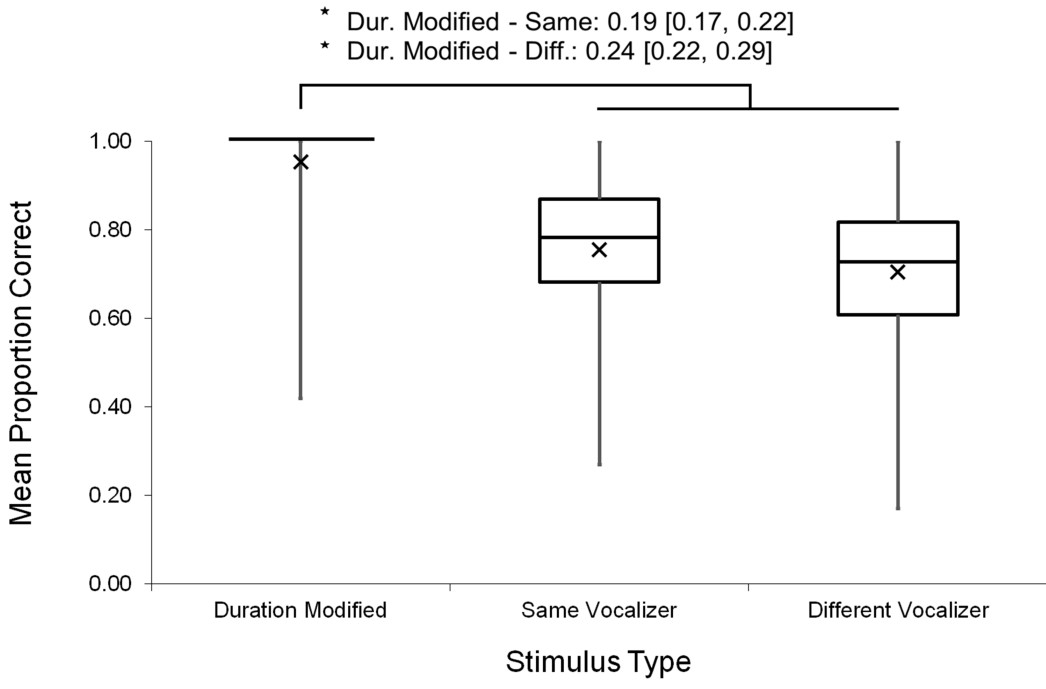

**Figure 2 Minimum, quartiles, and maximum response accuracies for Duration Modified, Same Vocalizer, and Different Vocalizer stimulus types; *X* represents group means.** Asterisk indicates significance, $p < 0.05$, with the following numbers indicating the estimate of the true difference of the means and the 95% confidence intervals.     

dimension (i.e., discriminated at chance level), and an empirical threshold of $d' > 1.00$ is conventionally used to indicate that reliable discrimination was achieved (*Macmillan & Creelman, 2004*; *Fugate, Gouzoules & Nygaard, 2008*). Overall, participants achieved $d'$ scores significantly above this criterion value ($t(103) = 11.02$, $p < 0.001$), indicating that they discriminated between trials in which they heard the same vocalizer and trials in which they heard two different vocalizers.

Mean response accuracies and latencies were further analyzed by stimulus type. One-sample *t*-tests indicated that, on average, response accuracies exceeded chance level (> 0.50) for each stimulus type (Duration Modified: $M = 0.95$, SE = 0.01, $t(103) = 38.76$, $p < 0.001$; Same Vocalizer: $M = 0.76$, SE = 0.01, $t(103) = 18.09$, $p < 0.001$; Different Vocalizer: $M = 0.71$, SE = 0.01, $t(103) = 14.15$, $p < 0.001$). A nonparametric Friedman test (used due to non-normal distribution of the Duration Modified data) with post hoc, Bonferroni-corrected, Wilcoxon signed-rank comparisons revealed that response accuracies varied significantly between stimulus types ($\chi^2(2) = 109.33$, $p < 0.001$; Fig. 2): as we had expected, participants discriminated more accurately on Duration Modified trials than on Same Vocalizer ($Z = 8.30$, $p < 0.001$) or Different Vocalizer trials ($Z = 7.83$, $p < 0.001$). Similarly, a repeated measures ANOVA revealed that mean response latencies varied by stimulus type ($F(2, 103) = 30.57$, $p < 0.001$, $\eta_P^2 = 0.229$; Fig. 3) such that, on trials in which participants responded correctly, they also reacted more quickly to Duration Modified trials than to Same Vocalizer ($-264.69 \pm 33.86$ ms, $p < 0.001$) or Different Vocalizer trials ($-235.57 \pm 36.95$ ms, $p < 0.001$). No significant difference was

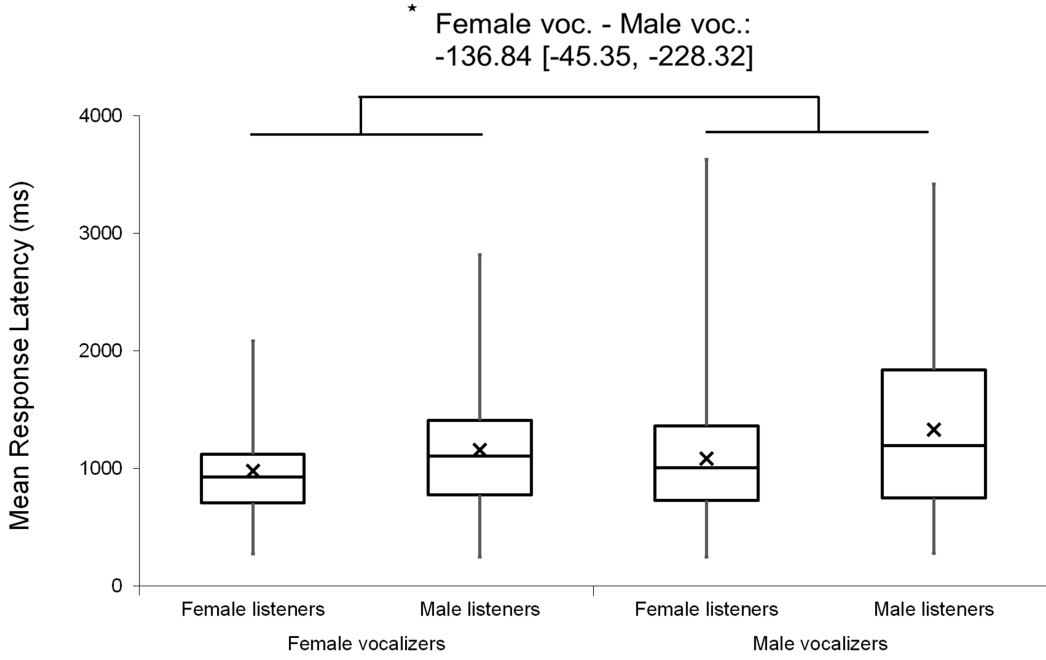

**Figure 3 Minimum, quartiles, and maximum response latencies on accurate responses for Duration Modified, Same Vocalizer, and Different Vocalizer stimulus types; X represents group means.** Asterisk indicates significance, $p < 0.05$, with the following numbers indicating the estimate of the true difference of the means and the 95% confidence intervals.

found between Same Vocalizer and Different Vocalizer trials in either response accuracies ($p = 0.035$; not significant with Bonferroni corrected $\alpha = 0.017$) or latencies ($p = 1.00$).

Altogether, these results suggest that, as we had anticipated, it was easier for participants to make accurate judgments on Duration Modified stimulus pairs (which, again, included a scream and a lengthened or shortened version of itself) than on either of the other stimulus types. In fact, 80 out of the 104 participants responded accurately on every Duration Modified stimulus pair, suggesting a ceiling effect. We therefore recalculated response accuracies and $d'$ scores excluding Duration Modified trials to determine whether listeners could discriminate vocalizers from unmodified screams. These calculations yielded a mean response accuracy = 0.73 (SE = 0.01) and a mean $d'$ score = 1.40 (SE = 0.06). Thus, although the overall discrimination level dropped when only Same Vocalizer and Different Vocalizer trials were taken into account, it remained significantly above the threshold value of 1.00 ($t(103) = 6.80$, $p < 0.001$), indicating that participants were able to discern whether unmodified scream exemplars where produced by the same individual or by two different individuals.

### Effects of listener and vocalizer gender

Mixed factor ANOVAs were used to explore possible main effects and interactions relating to listener and vocalizer gender. Vocalizer gender was used as a within-subjects factor and listener gender was used as a between-subjects factor, with $d'$ scores and mean response latencies used as output variables in two separate tests. The results revealed a small main effect of vocalizer gender on $d'$ scores ($F(1, 102) = 4.10$, $p = 0.046$, $\eta_p^2 = 0.039$),

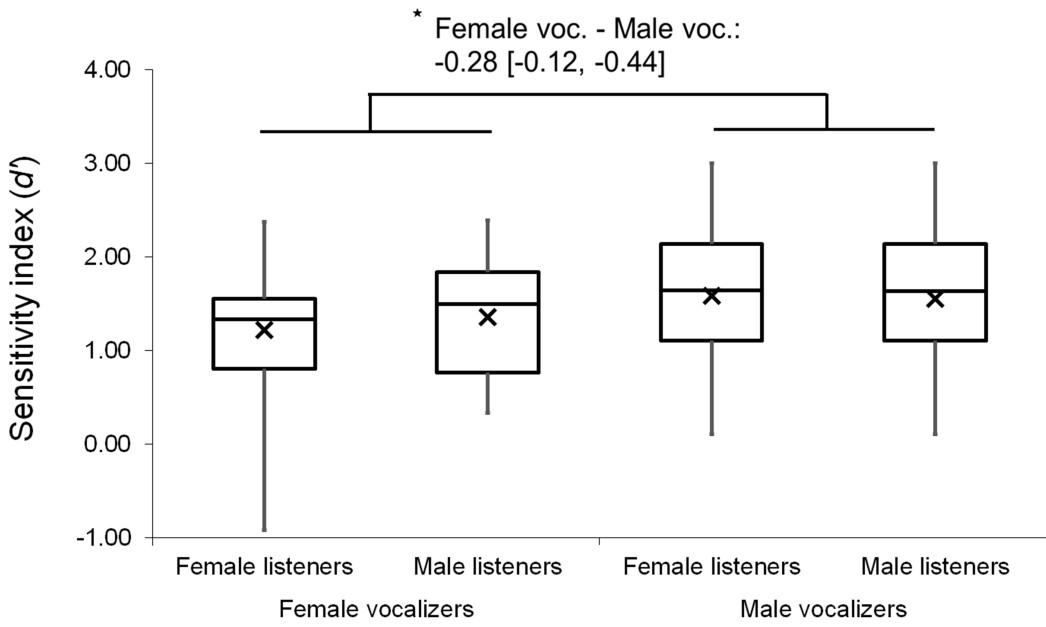

**Figure 4 Minimum, quartiles, and maximum sensitivity indices ($d'$ scores) by listener gender and vocalizer gender; $X$ represents group means.** Duration Modified stimuli were omitted for this graph. Asterisk indicates significance, $p < 0.05$, with the following numbers indicating the estimate of the true difference of the means and the 95% confidence intervals.

suggesting that listeners were better at discriminating male screams ($M = 1.72$, SE = 0.07) than female screams ($M = 1.55$, SE = 0.06). There was also a main effect of vocalizer gender on mean response latencies ($F(1, 102) = 16.18$, $p < 0.001$, $\eta_p^2 = 0.137$), indicating that listeners responded more slowly to male screams ($M = 1,120.63$, SE = 57.44) than to female screams ($M = 972.64$, SE = 43.93). No main effects of listener gender nor interacting effects of listener by vocalizer gender were found for $d'$ scores (listener gender: $p = 0.802$; interaction: $p = 0.492$) or response latencies (listener gender: $p = 0.064$; interaction: $p = 0.498$).

Analyses were repeated excluding Duration Modified trials, revealing a slightly larger main effect of vocalizer gender on $d'$ scores within this subset ($F(1, 102) = 11.57$, $p < 0.001$, $\eta_p^2 = 0.102$; Fig. 4), such that listeners discriminated more accurately when listening to male screams ($M = 1.57$, SE = 0.07) compared to female screams ($M = 1.26$, SE = 0.06). There also remained a significant effect of vocalizer gender on mean response latencies ($F(1, 102) = 8.80$, $p = 0.004$, $\eta_p^2 = 0.079$; Fig. 5) such that, when participants did respond accurately, they reacted more slowly to male screams ($M = 1,156.87$ ms, SE = 61.10) than to female screams ($M = 1,033.64$ ms, SE = 47.24). No significant main effects of listener gender nor interacting effects of listener by vocalizer gender were found for $d'$ scores (listener gender: $p = 0.641$; interaction: $p = 0.293$) or response latencies (listener gender: $p = 0.057$; interaction: $p = 0.467$).

### Effects of other participant attributes

Multiple linear regressions were conducted to explore the relationship between questionnaire-assessed participant attributes and their $d'$ scores. Predictor variables

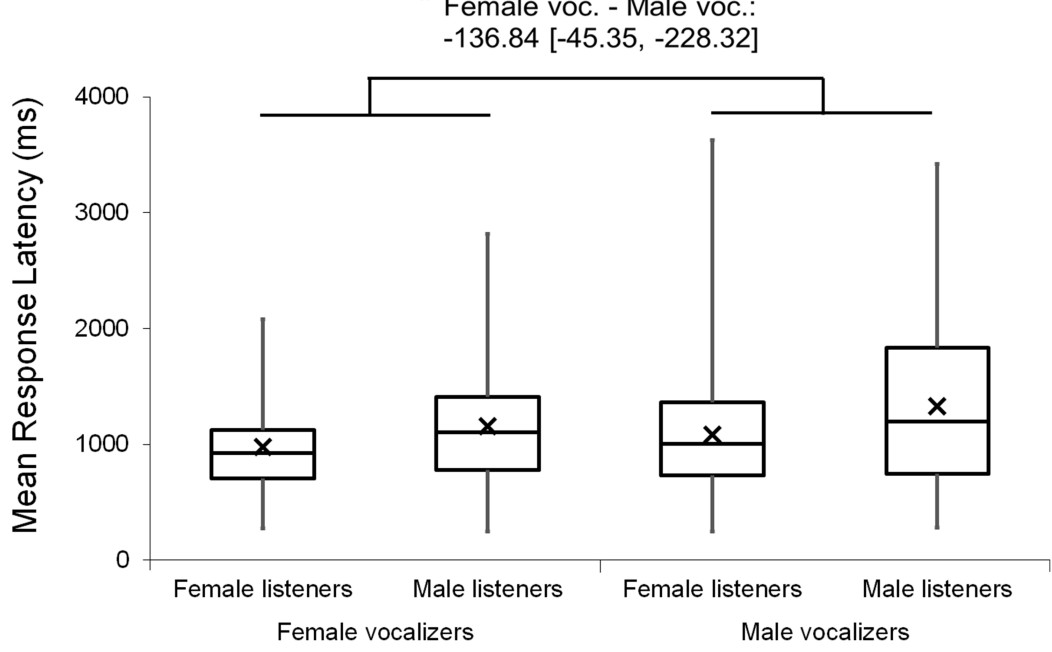

**Figure 5 Minimum, quartiles, and maximum response latencies by listener gender and vocalizer gender; X represents group means.** Duration Modified stimuli were omitted for this graph. Asterisk indicates significance, $p < 0.05$, with the following numbers indicating the estimate of the true difference of the means and the 95% confidence intervals.

included EQ and the four factors extracted from a PCA on questionnaire responses (Media, Games, Confidence, and Movies, Table 2). Two separate models were specified, one using the $d'$ scores calculated from responses across all trials, and one using the scores calculated with the exclusion of Duration Modified trials. However, neither model significantly fit the data better than null, intercept-only models (full dataset: $F(5, 98) = 1.90$, $p = 0.101$; reduced dataset: $F(5, 98) = 2.17$, $p = 0.063$).

Multiple linear regressions were also conducted to examine the relationship between the same predictor factors and mean reaction times on accurate responses in both the full dataset and the subset of data excluding Duration Modified trials. Again, neither model significantly fit the data better than null models (full dataset: $F(5, 98) = 1.38$, $p = 0.238$; reduced dataset: $F(5, 98) = 1.26$, $p = 0.288$).

## DISCUSSION

Our results suggest that human screams permit a level of individual discrimination, such that listeners could determine whether two screams were produced by the same vocalizer or by two different vocalizers. Listeners responded accurately especially on trials in which they were presented a scream along with its modified version (Duration Modified trials); however, even when responses on these trials were excluded—leaving only screams that varied naturally within and between vocalizers (Same Vocalizer and Different Vocalizer trials, respectively)—mean response accuracies and $d'$ scores significantly exceeded chance levels, suggesting that screams can convey sufficient information to discriminate individuals.

Previous research using isolated vowel sounds and laughter has shown that cues unique to speech per se are not required for the perception of vocal identity in humans (*Gaudrain et al., 2009*; *Kuwabara & Takagi, 1991*; *Lavan, Scott & McGettigan, 2016*; *Owren & Rendall, 2003*). Our results extend this work by demonstrating that identity-related vocal qualities are perceived in at least some screams as well. This finding is inconsistent with the results of *Hansen, Nandwana & Shokouhi (2017)*, who reported that listeners discriminated at chance levels in a same-different identity discrimination task using screams. That study used a very small sample size and did not account for potential heterogeneity in the data relating to participant ability or to the screams themselves. Our results are more in line with evidence that non-human primate screams, which share many of the acoustic characteristics of human screams, enable receivers to assess and respond to a caller's identity (*Cheney & Seyfarth, 1980*; *Fugate, Gouzoules & Nygaard, 2008*; *Seyfarth & Cheney, 2015*), suggesting that this informative capacity is one common to human and non-human screams.

We did not observe a main effect nor any interacting effects relating to listener gender. Our relatively small sample of male participants and vocalizers might have limited the power of these tests. However, the results did show significant variability relating to vocalizer gender: male screams were more accurately discriminated than female screams. Although this effect was marginal in the full dataset, it emerged clearly in the subset of data taken from Same Vocalizer and Different Vocalizer trials, that is, when the ceiling effect of Duration Modified trials was removed. We had predicted that gender effects might account for variation in this task because they also explain some variation in recognizing emotion from vocalizations (*Belin, Fillion-Bilodeau & Gosselin, 2008*; *Lausen & Schacht, 2018*). If anything, however, evidence suggests that females are better than males at expressing emotion through nonlinguistic vocalizations such as affect bursts (*Lausen & Schacht, 2018*), so it seems unlikely that the advantage conferred by male vocalizers here is related to enhanced emotional expression.

A possible alternative explanation for the vocalizer gender effect was proposed by *Owren & Rendall (2003)*, who also reported that listeners were better at discriminating identity when responding to male vs female exemplars, a finding they attributed to the lower fundamental frequencies (F0) and consequently richer harmonic spectra of male vocalizations. A similar proximate mechanism may be at work in this study, but we note that findings regarding sex differences in vocal communication merit consideration of evolutionary issues as well. Screams are evidently more widely produced by females than by males (*Anikin & Persson, 2017*), a tendency we share with some nonhuman primates (*Bernstein & Ehardt, 1985*). One possibility, then, given the relative rarity of male screams, is that they differ from female screams with respect to salience to listeners, a quality that in turn could affect listeners' interpretations. Although it is difficult to evaluate this explanation here, particularly because female screams were discriminated more quickly than male screams, further exploration of sex and gender differences in screams is likely to prove a fruitful avenue of research.

We did not find strong evidence in analyses of either accuracies or response times to support the hypothesis that prior experience with screams would facilitate the perception

and use of identity cues. It is possible, instead, that experience with a specific individual's vocal qualities, rather than experience with the vocal class as a whole, would enable better discrimination from that individual's screams, as has been demonstrated in laughter (*Lavan, Scott & McGettigan, 2016*) and speech (*Schmidt-Nielsen & Stern, 1985*). We also note that our questionnaire only assessed experience with screams in the media. Future research might therefore explore broader measures of experience with screams, including those relating to personal familiarity with the vocalizers as well as exposure to screams in real-life scenarios.

Additionally, we did not find evidence that differences in empathy accounted for variation in participants' abilities to discriminate identity. Although empathy likely plays a role in motivating the decision to respond to screams (*De Waal, 2008*), it is plausibly less involved at the stage of identity perception. Given that our hypothesis with respect to listener gender was also not borne out—any gender differences in emotional processing did not translate into gender differences on this task—these results may suggest that the recognition of identity is a process somewhat independent from that of recognizing or experiencing the emotional aspects of screams, consistent with models of voice perception that posit separate functional pathways for the processing of vocal identity and affect (*Belin, Fecteau & Bedard, 2004*). More research is needed, however, to test this hypothesis specifically in screams.

*Owren & Rendall (2003)* contended that screams are not ideal for communicating identity in part because they lack the stable, tightly-spaced harmonics of some other vocalizations. They argued that the distribution of energy across a rich, tonal harmonic spectrum is important for revealing the individually distinctive effects of supralaryngeal filtering. Consistent with this claim, a same-different identity discrimination task in their study using the spoken vowel sound /ɛ/—the acoustic profile of which is marked by closely stacked harmonics—yielded a mean $d'$ score of 3.5 (*Owren & Rendall, 2003*), much higher than the average scores observed in our experiment. However, increasing evidence suggests that vocal cues to identity in humans are redundant (*Mathias & Von Kriegstein, 2014*). That supralaryngeal filter effects seem important for individual recognition from tonal vocalizations does not necessarily mean that vocalizations where filter effects are masked will lack cues to identity. Indeed, humans can recognize vocalizers from speech even when all harmonics except F0 (a parameter related to source vibrations rather than the filter) have been eliminated (*Abberton & Fourcin, 1978*), and research in other nonlinguistic vocalizations suggests that F0 as well as spectral characteristics may vary systematically between individuals (e.g., in infant cries; *Gustafson et al., 2013*). Another intriguing possibility in screams is that nonlinear phenomena such as chaotic noise and bifurcations between harmonic regimes might serve as additional identity cues (*Fitch, Neubauer & Herzel, 2002*). The role of these acoustic parameters in communicating identity in screams is a promising question for future research.

One issue that has received increased attention in recent literature is that of potential differences between acted and naturally occurring nonlinguistic vocalizations (*Anikin & Lima, 2017*; *Anikin & Persson, 2017*; *Bryant et al., 2018*; *Engelberg & Gouzoules, 2018*). Of particular relevance here, *Lavan et al. (2018b)* found that listeners' abilities to discriminate identity from laughter varied as a function of vocal production mode,

that is, whether the laughter was spontaneous or volitional. In that study, participants' discrimination abilities were significantly impaired when presented with spontaneous laughter; indeed, their $d'$ scores fell below those obtained from the present experiment. Given that most screams in our study were produced by professional actors, these findings in laughter raise the possibility that our participants may also have discriminated less accurately had we incorporated more naturally produced screams. For example, acted screams are possibly more stereotyped than natural ones within an individual, which could facilitate the matching of two exemplars produced by the same vocalizer. That said, our previous study regarding this issue showed that actors are capable of producing credible screams (*Engelberg & Gouzoules, 2018*). An additional possibility is that the lower $d'$ scores for spontaneous laughter relative to screams reflects the inherent differences of two distinct vocal classes. Laughter is a highly acoustically variable vocalization (*Bachorowski, Smoski & Owren, 2001*), which may hinder listeners' abilities to recognize an individual. We have begun to characterize the acoustic variation within screams (*Schwartz, Engelberg & Gouzoules, in press*), but the full extent of this variation within individuals remains to be established.

Potential confounds in our experiment might include any acoustic variation in Different Vocalizer pairs introduced from sources other than individual identity, such as differences in the emotional contexts of screams or the age of vocalizers. Although we matched both of these factors within stimulus pairs to the fullest extent possible, exact matches (e.g., for the precise age of every vocalizer) were not always feasible. Conceivably, slight differences in vocalizer age might have contributed to listeners' judgments, as the mean age difference in Different Vocalizer pairs would exceed that of Same Vocalizer pairs. That said, we know of no empirical evidence in the literature to suggest that listeners can assess age from screams, especially with the degree of precision necessary to steer participants' responses on our task. Likewise, although our sample size precluded context-based analyses, it would be of interest for future research to account for the role of emotional context in identity perception from screams, particularly because screams are so contextually heterogeneous (spanning contexts of fear, anger, joy, and pain; *Anikin & Persson, 2017*). One open question concerns the extent to which identity cues are context-specific as opposed to consistent across contextually disparate screams. Ongoing and planned investigations in our lab, exploring the acoustic and perceptual differences between screams produced in different emotional contexts, may shed some light on these issues.

A final potential if unlikely confound is the possibility of learning effects. Given that 21 exemplars and 15 vocalizers appeared in multiple stimulus pairs, listeners may have benefited from hearing the same scream more than once across the experiment. However, we suggest that if any such benefits occurred, listeners would likely have learned cues distinctive to the repeated vocalizers, consistent with the conclusion that screams convey identity cues.

It is important to note a distinction between the discrimination tested here and true individual recognition. Our study required that participants listen to two consecutively presented exemplars and decide whether they were produced by the same vocalizer or different vocalizers; recognition entails matching a perceived vocalization against a stored,

cognitive representation of a familiar individual's vocal attributes (*Sidtis & Kreiman, 2012*; *Steiger & Müller, 2008*; *Tibbetts & Dale, 2007*). When a non-human primate in a playback experiment responds preferentially to her offspring's screams (*Cheney & Seyfarth, 1980*; *Gouzoules, Gouzoules & Marler, 1986*), she is likely demonstrating recognition at some level, especially because many such studies present listeners only one call or short bout within a span of several days (*Cheney & Seyfarth, 1980*). Our results demonstrate a prerequisite to this kind of vocal recognition, in that human screams are sufficiently distinct between vocalizers to permit immediate discrimination, but further research is needed to determine whether listeners could identify a scream as belonging to a particular individual. Additional questions concern the mechanisms underlying individual discrimination from screams and the processes involved in the development of these abilities.

## CONCLUSIONS

The finding that human screams convey cues to screamer identity is consistent with the wealth of evidence for a similar capacity in nonhuman primates (*Bergman et al., 2003*; *Cheney & Seyfarth, 1980*; *Fugate, Gouzoules & Nygaard, 2008*; *Gouzoules, Gouzoules & Marler, 1986*; *Kojima, Izumi & Ceugniet, 2003*; *Seyfarth & Cheney, 2015*; *Slocombe et al., 2010*). Nonetheless, this question has been the subject of considerable and ongoing debate (*Fugate, Gouzoules & Nygaard, 2008*; *Owren & Rendall, 2003*; *Rendall, Owren & Rodman, 1998*). The present study contributes to this area of research by demonstrating an ability among humans to discriminate between individuals based on their screams. This result supports evolutionary continuity in at least some aspects of scream function between humans and other primate species. In general, more is known about the functions of screams in nonhuman primates than in humans; more research is needed in order to elucidate which aspects of human scream production and perception are evolutionarily shared and which are derived.

## ACKNOWLEDGEMENTS

We thank Caitlin Clark, Alexander Gouzoules, Leah Friedman, Elizabeth Harlan, and NooRee Lee for assistance with stimulus collection, as well as Anna Duncan for assistance with data collection.

### Funding

Jay Schwartz was supported by the National Science Foundation Graduate Research Fellowship under Grant No. DGE—1343012. The funders had no role in study design, data collection and analysis, decision to publish, or preparation of the manuscript.

### Grant Disclosures

The following grant information was disclosed by the authors:
National Science Foundation Graduate Research Fellowship under Grant No: DGE—1343012.

## Competing Interests

The authors declare that they have no competing interests.

## Author Contributions

- Jonathan W. M. Engelberg analyzed the data, prepared figures and/or tables, authored or reviewed drafts of the paper, approved the final draft.
- Jay W. Schwartz analyzed the data, prepared figures and/or tables, authored or reviewed drafts of the paper, approved the final draft.
- Harold Gouzoules conceived and designed the experiments, performed the experiments, authored or reviewed drafts of the paper, approved the final draft.

## Human Ethics

The following information was supplied relating to ethical approvals (i.e., approving body and any reference numbers):

This research was conducted in compliance with Emory's Institutional Review Board under IRB00051516, approved July 26, 2011.

## Data Availability

The raw data are available in Data S1. The data show all participant information (collected originally on paper and transferred to the data file) and their responses and reaction times on every trial (collected by E-Prime 2.0 software). The stimuli used in the experiment are provided in Audio S1. The questionnaire developed and used in the experiment is provided in Supplemental Materials S1.

## Supplemental Information

Supplemental information for this article can be found online at http://dx.doi.org/10.7717/peerj.7087#supplemental-information.

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
