# Peer review of "Do human screams permit individual recognition?"

_PeerJ, doi:10.7717/peerj.7087_

## Round 0.1 · original submission · Major Revisions

Each reviewer offered a detailed and balanced critique of your paper that suggests the research is a potentially strong contribution to the literature. I agree the manuscript had promise and the topic is interesting. Too often, non-human primates are the focus of sensory perception work because the assumption is that humans have removed themselves from natural environments. Of course, we retain our sensibilities and studies such as yours which examine perceptual qualities of sensory outputs in humans are highly valuable for understanding our place in nature and our own evolutionary path. Thus, while I have decided the manuscript requires major revision, I strongly encourage you to resubmit a new version with the additional information and increased clarity requested by your peer review.

Reviewer 1 notes that data from related laughter studies suggest that the context of production may influence listener perception. Generating new data to address this point is beyond the scope of the current study but should be discussed as appropriate. Reviewer 2 noted that acted versus spontaneous vocalizations have potentially different qualities which might influence recognition. The acted vocalization is not discussed as a potential confounding factor and should be considered. Reviewer 3 raises several points regarding improving the discussion to include greater detail on all analyses conducted. R3 also notes that there is not a strong enough link between the introduction that frames the problem and the discussion which provides your contextual understanding of study’s results. All reviewers had additional questions about methods and results that should be addressed for both clarification and replicability. While I realize that your data precluded analytically engaging with Reviewer 1’s comments on vocal properties, the point raised is intriguing and merits discussion; specifically, R1 notes that perhaps the inherent vocal properties as measured by standard tools may indicate acoustic properties that are transmitted across all types of vocalizations. And, R3 discusses quantitative methods of analyzing the vocalizations.

All reviewers offer additional suggestions for improving the paper and I strongly encourage you to consider their comments in the revision. I realize that, in a few instances, the information requested by reviewers is present in the paper. In these cases, consider highlighting that information in some way—shortening a paragraph or re-organizing text. I find that when important information is missed, that information is likely buried with other less important information.

I would like more information about the validation of the survey created for the study—R3 comments on the survey tools as well. I do not see any information validating those data and wonder how accurate a tool it is, considering it plays a role in the down-stream analysis. Is the in-house survey tool meant to complement the validated empathy tool? I am troubled the re-analysis of the d’ scores. The Friedman test is an appropriate tool for these data and identified significant differences across groups (Same-A as different). The only reason for excluding the Same-A scores in a second analysis is to find the desired difference between Same-B and Different—a difference that makes the paper more compelling because, as you note, we would expect most listeners to identify a scream and its modified version as the same. The real interest lies in how well people are able to match screams to the same person when taken from different bouts or different contexts. The GLM suggested by R1 might help with this. If you do stand by your current secondary analysis, you will need to justify this as statistically appropriate which will be hard to do because you are essentially rejecting the validity of the Friedman test. Ultimately, this might be an issue of sample size. The power, were this an ANOVA, would be very low for a sample of 104 individuals. I am a bit confused by F1-F4. You reference them in Line 362 as F1-F4, supplemental materials (but should be Supplemental Table 1) without further explanation. I am not sure ST1 should be supplemental since it is a result that is used for further analysis. So, I would recommend moving it into the body of the paper. Moreover, help the reader and simply stated what these four elements are in text and please introduce a short-hand phrase that can be used a reminder in-text for each and/or as a footnote in tables. As is, the tables aren’t very meaningful without extra work on the part of the reader. Finally, please also include the survey and the sound bites (or an explanation as to why you aren’t sharing those data which could be used to replicate the study. A table with detailed information on the company, number of sounds (even if just screams), and hyperlink would be useful for the commercially available sound banks (referencing text in paragraph starting with line 160).

·

Basic reporting

Engelberg et al. report an experiment exploring whether human listeners can accurately perceive identity from (human) screams through a speaker discrimination task. The authors report that screams can indeed convey information about speaker identity (d’ > 1 and mean accuracy > .5), with some effects of exposure, confidence and vocaliser gender are also apparent. An acoustic analysis did not yield any significant effects.

The study is written and structured in a clear way. The literature on the topic is reviewed, referenced and synthesised into an informative introduction section. Figures and tables are mostly adequately labelled and the raw data has been shared. Hypotheses and predictions are started clearly at the start of the introduction. Some minor comments about the basic reporting are listed below:

1. The authors state that little research into identity perception from non-verbal vocalisations, such as laughter and screams has been conducted for humans (l. 66). I would point out that there we have run a couple studies looking at identity perception from laughter, which may be relevant to the current study, especially given that there are only a small number of studies available.

-Lavan N., Scott S.K. & McGettigan C. (2016). Impaired generalization of speaker identity in familiar and unfamiliar voices. Journal of Experimental Psychology: General, 145(12), 1604-1614.
- Lavan N., Short B., Wilding A., & McGettigan C. (2018). Impoverished encoding of speaker identity in spontaneous laughter. Evolution and Human Behavior, 39(1), 139-145.

2. The x axis in Figure 1 is labelled as showing percent but the number only range from 0 to 1 - please amend this.

3. For all figures, it would be nice to explain in the figure legends what the error bars are showing.

Experimental design

The study falls within the aims and scope of articles published within PeerJ. As the authors accurately point out, little is known about the perception of identity from nonverbal/nonlinguistic vocal signals in humans and this study thus provides a novel contribution, addressing a gap in the literature.

The experimental design appears to be well thought out and appears to have been well-conducted. Methods are described adequately and in sufficient detail - I have some comments listed below regarding the some of the details of the design and the statistical analysis.

1. The current study includes a wide range of screams that do not only differ in their broad acoustic properties but seemingly also in the contexts in which they were produced. Since there is previous evidence showing differences in the perception of a type vocalisation (e.g. laughter) produced in different contexts, do the authors think that similar effects could be present for screams? It is of course difficult to post-hoc reconstruct why a scream was produced and the number of screams was limited in this study which precludes any formal analysis. It might however benefit the paper to include some notes on the heterogeneity of the types of screams - not only in terms of their acoustics but also the context they were produced (pain, anger, fear, etc.) in the discussion of the paper.

2. I was wondering whether the items within the stimulus pairs were randomised across participants or whether all participants were presented with the same pairs? From what I glean from the methods, it sounds like all participants were presented with the same pairs.

3. While I will not insist on this being changed for this manuscript, I would strongly encourage the authors to consider running their statistical analyses using (generalised) linear mixed models. This would allow them to account for participant as well as item-specific effects. Accounting for the item-specific (or here pair-specific) effects seems to be particularly important in the current study if the pairs were not shuffled (see point 2 above), thus resulting in a relatively low number of different pairs.

4. I was also finding myself getting a little confused when the for sections of the results, analyses were reported with the SAME-A trials included and then excluded, alongside analyses including and excluding the accidentally duplicated pairs as well as with and without premature responses. While I appreciate that some of these analyses are reported to provide full transparency, it might be desirable to streamline the reporting of the results a little bit. This could be achieved by perhaps always excluding the SAME-A trials after showing that they differ from the remaining trials (analyses could be moved into supplementary materials) and consistently omitting duplicated items and pre-mature trials.

5. While it is commendable to assess participants everyday exposure to screams via a questionnaire, this questionnaire is only tackles exposure through media while neglecting real-life exposure to screams. I can appreciate why this was done - perhaps to avoid asking questions that may be ethically a bit challenging - a broader measure of experience would have been interesting. A brief acknowledgement that screams are also encountered outside of video games might be useful in the discussion or methods section.

6. I was curious about why this particular type of acoustic analysis was chosen. While I am aware that it can be challenging to extract reliable acoustic measures for, for example, mean/median F0, spectral measures or measures of periodicity from some non-verbal vocalisations, I am from my reading to the literature on this topic (in psychology) a bit more used to seeing this type of analysis for human vocalisations. Maybe an analyses using these types of measures might shed some more light into the acoustic properties that may underpin listeners’ judgements?

Validity of the findings

The sample size (>100 participants) is impressive although some effects of listener and speaker characteristics are small and seem bounce between being statistically significant and not significant depending on which trials are excluded or included (see point 4 of the previous section). The effects relating to the main research question appear to be however robust and sizeable.

Much like the introduction, the discussion is well structured and clearly written. Results are linked back to and evaluated within the existing literature.

Additional comments

All in all, I believe that this study is well-conducted and of interest to the community and should be published after some revisions to mainly tidy up the reporting of the results.

·

Basic reporting

The authors examine the ability of human listeners to gauge individual identity from screams. This paper represents an important, and timely, contribution to the very small (but growing) body of research on human nonverbal vocalisations.

The paper is well written – clear, succinct, and with a logical flow. However the Introduction/Discussion sections would benefit from the addition of several omitted yet highly relevant references, as noted below.

Perhaps the most important references omitted are a series of recent papers examining identity perception from laughs (and other vocal stimuli) from Lavan et al. ( Scott/ McGettigan labs).

For instance, Lavan et al., 2018c (see refs below) have shown a limited capacity, in humans, to identify individuals from *spontaneous* laughter. These papers are obviously highly relevant and should be discussed (currently only one is cited) – particularly the potential distinction between cues to identity in volitional vs spontaneous nonverbal vocalisations. Based on the sources for stimulus extraction outlined by the authors (L 160), it seems most scream stimuli used in this paper were probably ‘acted’, e.g., from films and adverts, and therefore would not classify as *spontaneous*. However, some of the screams could have been genuine (e.g., those taken from You Tube videos or sound banks). The high degree of accuracy in listeners’ ability to gauge identity from screams (current paper) compared to laughter (Lavan et al.) might be due in part to the use of acted versus genuine vocal stimuli. Acted screams could for instance follow a more stereotyped pattern within individuals and thus be easier for listeners to match. Another (non-mutually exclusive) explanation for the different results of these papers could be the type of vocalisation – e.g., laughs are lower-pitched and likely to vary more, acoustically, across bouts within-individuals compared to screams.

Other recent and highly relevant papers on human nonverbal vocalisations (incl. distress [fear and pain] screams) that have been omitted include:
- Raine et al., 2017, 2018a, b (recognition of sex, strength, body size etc. from nonverbal vocalisations);
- Anikin & Lima, 2017 (Perceptual and acoustic differences between authentic and acted nonverbal vocalizations)
- Anikin & Persson, 2017 (Corpus of naturalistic nonverbal vocalisations from amateur videos)

The authors could also include a short discussion of the relevance of nonverbal vocalisations, from an evolutionary and social (functional) perspective, and highlight the important ways in which nonverbal vocalisations differ from speech in this regard (acoustically/functionally). Nonverbal vocalisations are a common part of the human vocal repertoire yet hugely understudied. This fact underscores the importance of this paper, and calls for more research in this area.

Can the authors include the 10-item questionnaire, developed in their lab to measure exposure to screams in media, as supplementary material?

Including some, if not all, the scream pairs as supplementary material (especially those corresponding to the spectrograms in Fig 5) would also increase the transparency and replicability of this work.

The green and blue colours used for bar graphs are of a similar intensity and will not reproduce well in black and white. I suggest using more contrasting hues.

Experimental design

In general, the methods are clear and fairly well justified, particularly stimulus selection, stimulus preparation and playback procedures.

Where the methods are lacking is in acoustic analysis.

The acoustic analysis section (L277 – 301) is vague and poorly justified. For example, “Spectrograms of screams were generated in Adobe Audition CC and qualitatively inspected in comparison to Same Scores. Based on these initial observations, we identified two acoustic variables that seemed to correlate with participant responses: harmonic regime and bifurcations.”
- What exactly do the authors mean by “we identified two acoustic variables that seemed to correlate with participant responses”?
- Which acoustic features did the authors consider in their analyses, initially?
o I am surprised that fundamental frequency/pitch (F0) is ignored, as this is a key predictor of vocal distress and probably the most salient nonverbal feature of human vocalisations. It is also a highly sexually dimorphic feature of the voice that could partly explain the sex-effects observed, e.g., 336. Lower-pitched (male) vocalisations will have a denser harmonic spectrum that could assist in formant discrimination and identity cues as noted on L 425 (see also Charlton et al., 2013; Pisanski et al., 2014). In examining F0 the authors could test the prediction noted on L445, as well as how F0 predicted listeners’ judgments/accuracy scores.
o Formants, as a key identity cue, could also be examined – though formants may be too difficult to measure from very high F0 vocalisations (as in Raine et al. 2017 with tennis grunts).
- Were observers (L 285) blind? What criteria did they rely on to make these classifications?
- Why was duration used for the Same-A condition, rather than e.g., F0?

On what basis were sample sizes (of voices/listeners) selected?

The uneven ratio of males to females (more females), both in vocal stimuli (Table 1) and among raters, should be discussed as a limitation. While an overrepresentation of women is common in student samples from psychology/behavioural sciences departments, the overrepresentation of female voices in the scream stimuli is more difficult to justify. How was this biased ratio controlled in the analyses of gender effects (L 336)?

Please also note whether the vocalisations were fully randomised, or blocked by sex, and how this might have affected accuracy in identification rates (considering also the biased sex ratio for screams).

Using a PCA on questionnaire items is not very common. Can the authors explain the rationale for this approach in more detail and provide justification (e.g., references?) for its validity?

Validity of the findings

Screams were extracted from online sources such as films and sound banks, from which three types of stimulus pairs were composed: Same-A (same scream from same vocaliser, with two different durations); Same-B (different screams from same vocaliser); Different (different screams from different vocalisers). Listeners were then tasked with indicating whether the two screams in each pair were produced by the same person, or two different people. The results indicate an impressive level of accuracy in listeners’ ability to gauge identity from screams, with higher than 70% accuracy scores in all three conditions.

As described in Section 1 (Basic reporting), a key issue that is largely ignored in this paper is the distinction between spontaneous/genuine vs. volitional/acted vocalisations, as identity cues may be more salient in the latter. The authors should explicitly comment on the proportion of scream stimuli that were likely acted vs genuine (to the extent that this can be discerned) and, most importantly, how this could affect their results. If possible, I would suggest controlling for this factor in their analyses. I’ve listed many references relevant to this topic below.

In terms of acoustic analysis, the authors conducted a post-hoc investgiation, and only examined harmonic regime and bifurcations. As described in Section 2 (Experimental design) I strongly suggest including additional justification for, and description of, their acoustic analyses, and to expand on the number of acoustic features analysed.
It is unclear why the authors have omitted several key acoustic factors (particularly F0, formants, and duration) from their analyses, given evidence that these features are key predictors in listeners’ voice-based judgments of a variety of speaker traits (and F0/formants are highly sexually dimorphic). Formants are also a key identity cue in humans, as in other mammals – see e.g., references listed below. Duration and F0 are also important predictors of genuine vs. authentic vocalisations (see e.g., Bryant & Aktipis 2014; Anikin & Lima, 2017).


References :

Anikin, A., & Persson, T. (2017). Nonlinguistic vocalizations from online amateur videos for emotion research: A validated corpus. Behavior research methods, 49(2), 758-771.

Anikin, A., & Lima, C. F. (2017). Perceptual and acoustic differences between authentic and acted nonverbal emotional vocalizations. The Quarterly Journal of Experimental Psychology, 1-21.

Bryant, G. A., & Aktipis, C. A. (2014). The animal nature of spontaneous human laughter. Evolution and Human Behavior, 35(4), 327-335.

Charlton, B. D., Taylor, A. M., & Reby, D. (2013). Are men better than women at acoustic size judgements?. Biology letters, 9(4), 20130270.

Lavan, N., Burton, A. M., Scott, S. K., & McGettigan, C. (2018a). Flexible voices: identity perception from variable vocal signals. Psychonomic Bulletin & Review, https://doi.org/10.3758/s13423-018-1497-7
Lavan, N., Domone, A., Fisher, B., Kenigzstein, N., Scott, S. K., & McGettigan, C. (2018b). Speaker Sex Perception from Spontaneous and Volitional Nonverbal Vocalizations. Journal of Nonverbal Behavior, 1-22.

Lavan, N., Short, B., Wilding, A., & McGettigan, C. (2018c). Impoverished encoding of speaker identity in spontaneous laughter. Evolution and Human Behavior, 39(1), 139-145.

Lavan, N., & McGettigan, C. (2017). Increased discriminability of authenticity from multimodal laughter is driven by auditory information. The Quarterly Journal of Experimental Psychology, 70(10), 2159-2168.

Pisanski, K., Fraccaro, P. J., Tigue, C. C., O'connor, J. J., & Feinberg, D. R. (2014). Return to Oz: Voice pitch facilitates assessments of men’s body size. Journal of Experimental Psychology: Human Perception and Performance, 40(4), 1316.

Raine, J., Pisanski, K., & Reby, D. (2017). Tennis grunts communicate acoustic cues to sex and contest outcome. Animal Behaviour, 130, 47-55.

Raine, J., Pisanski, K., Oleszkiewicz, A., Simner, J., & Reby, D. (2018). Human listeners can accurately judge strength and height relative to self from aggressive roars and speech. iScience, 4, 273-280.

Raine, J., Pisanski, K., Simner, J., & Reby, D. (2018). Vocal communication of simulated pain. Bioacoustics, 1-23.

Additional comments

Please refer to comments from Sections 1-3 above.

Reviewer 3 ·

Basic reporting

The manuscript is largely well-written, appropriately structured and unambiguous. All of the relevant raw data and analysis outputs were made available. Overall, the study will be a worthwhile addition to the literature, but in my assessment there are several issues with the analysis and Discussion that should be addressed before publication.

Please see below for specific comments on these matters.

Experimental design

I found the paradigm used by the authors to be appropriate for examining the extent to which humans are able to identify whether sequentially heard screams were produced by the same vocaliser. The ecological validity of this paradigm to the 'real world' problem of being able to identify which of 30 odd group-members produced an unseen scream (as demonstrated in primate literature) is debatable, but this is acknowledged by the authors in their Discussion. However, it would greatly improve the Introduction if these limitations were also outlined there, as well as how the specific aims of this study will inform the broader question. Currently, this link is not as clear as it could be.

The methods were very clearly and thoroughly described such that replication would be straightforward. However, the statistical software used for analysis is not specified, which I would encourage. If R was used, then the relevant scripts would be a welcome addition to the supplemental materials.

My biggest methodological concern is that the authors approach to acoustic analysis was somewhat unconventional. They have taken a qualitative (two researchers deciding which screams are similar), rather than quantitative (directly measuring acoustic parameters to determine which are similar), approach towards determining similarity between screams. I think the manuscript would greatly benefit from a justification and cited precedent for the methods used here, or else a more formal acoustic analysis.

Validity of the findings

The primary findings regarding discrimination of screams are convincing and appropriately analysed. The authors' conclusions regarding this are grounded and well-justified.

Regarding the more tertiary analyses (response latency, the questionnaire and gender effects) the manuscript is considerably weaker, largely owing to the fact that the Discussion is currently quite lightweight. It could be strengthened by elaboration on the broader implications of individual findings, placing them within the context of the broader literature. The authors performed quite a thorough analysis of many factors, but many of the findings are glossed over, particularly with regards to the questionnaire data. For example, analysis of the Empathy Quotient is not mentioned in the Discussion at all, despite the lack of association with accurately discriminating screams contrasting somewhat with prior research mentioned in the Introduction. In general, if a variable was worthy of analysis then it should also be worthy of discussion even if it did not yield statistical significance.

In general, the questionnaire portion of the study feels somewhat extraneous, largely owing to the fact that discussion of the implications of both positive and negative findings relating to it are scarce. For example, the authors report that increased exposure to video-games was associated with accuracy, but then quickly write it off as likely being a consequence of the fact that gamers have experience of responding rapidly to computerised stimuli. If even positive results on items like this are of no significance to the underlying research question, then it is unclear why they were included in the first place.

Furthermore, the rational for examining gender effects is raised in the Introduction, but the findings on this (male voices are classified with greater accuracy, but female voices more quickly) are not discussed within the context of this literature. On this note, the argument regarding speed/accuracy trade-off with regards to male and female vocalisers is not very clearly expressed. I am not sure what the authors mean by males being easier to accurately identify due to but females being more ‘perceptually salient’, or why this would result in the trade-off mentioned.

I have recommendations for several aspects of the Results section:

Response latencies were calculated, but only from correct trials. If the authors have a good rational for this then it is not apparent in the text. Comparing latencies between correct and incorrect answers may shed further light on the ‘speed/accuracy trade-off’ discussed later in the paper.

In lines 343-344 the authors report whether the outcome of their analysis on gender effects is different if premature responses (given before the onset of the second stimulus) were included in the dataset. It is unclear why was done, since these responses cannot possibly provide any information on scream recognition. My preference would be to simply report what proportion of responses were given prematurely and throw this data out.

Finally, I recommend the use of boxplots (or similar) rather than bar-charts for all figures as they are much more effective at presenting the distribution of responses in a dataset.

Additional comments

Below are a number of minor comments and suggestions which I would also recommend the authors address but are not critical.

Line 78: remove the words ’serve to’ so that it simply reads ‘elicit’. I am also not sure this matches definitions of altruism within literature on the topic – helping behaviour/assistance could be more appropriate?

Line 131: What is an ‘inherently stable vocalisation’?

Line 188: I do not understand why the gap is ‘roughly’ 2000ms. Surely this would be under precise control when constructing the stimuli? If there is a range of values for some reason, the authors should clarify the reason and provide the range and average gap.

Throughout the Methods and Results, where a t-test or similar is used, the text should specify whether it was one- or two-tailed.

Lines 293-294: I’m not sure that this is best practice. The purpose of inter-observer reliability is surely to check for independent agreement between two observers; if they then set about with the aim of reaching absolute agreement on each item then this undermines that goal. This could be made more transparent by reporting how many screams were categorised in this fashion.

Line 420: ‘consistent main effect’ is not very clear phrasing. I assume the authors mean that there was no significant effect.

Line 322: This should specify again that only correct trials were examined

Line 324: This should specify the alpha-level after bonferonni correction

347-348: Statistical outputs should be included for the final claim in this paragraph.

Line 351: Throughout the manuscript, the authors should replace instances where ‘better’ performance is referred to with something more specific (‘more accurate’, etc). It is currently not clear until halfway through this paragraph whether the opening sentences refers to accuracy or latency.

Line 357-358: here a ‘small effect’ is referred to but no effect size is given. A marginally significant p value is not the same thing as a small effect.

387-388: The phrase ‘approaching statistical significance’ should always be avoided as it is statistically nonsensical, particularly when the value is as distant from the alpha-level as 0.097

Line 399: ‘permit’ is an unusual word choice here, as though screams themselves rather than listeners are the agents in this sentence. This also applies to the title of the manuscript.

Line 401: As discussed in a prior comment, replace phrases like ‘performed well’ throughout the manuscript with a specific outcome.

Line 435: Did measures of confidence also correlate with response latency?

Line 439-440: The fact that F2 was no longer significant when Same-A was included suggests a ceiling effect in this condition (which we can see in figure 1).

---

## Round 0.2 · Minor Revisions

All three reviewers feel the revision is excellent and has improved their understanding of methods, analysis, and interpretation of results. There is one major lingering issue, the acoustic analysis. Two of the expert reviewers have raised the issue that this revised version has not improved the justifications for including the analysis with the limited number of seemingly less informative variables. I tend to agree that either a more traditional and comprehensive approach be used as suggested in the reviewer comments or the analysis be removed. The paper is very interesting and all three reviewers have engaged with interest in the ideas and findings, which is highly promising for a second paper on the acoustic analysis (if omitted from a resubmission). As the analysis was done as a post hoc test, I do not believe its exclusion will hinder the value of the paper.

There are a few minor additional revisions requested by one reviewer regarding confounding factors to the interpretation of the results. Please address those in a short sentence where appropriate to improve the discussion and identify ways forward for other researchers.

·

Basic reporting

No comment

Experimental design

No comment

Validity of the findings

No comment

Additional comments

The authors have addressed most of my concerns to my satisfaction - thank you!

I would only add that given that all 3 reviewers seemed to agree that the acoustic analysis is somewhat unconventional, doing at least a basic analysis of F0 and - if possible - formant frequencies would benefit the paper as other readers will probably have similar questions/concerns to the current set of reviewers. This type analysis seems to be no more fleshed-out than the original one and it would just be an alternative (that could be put into supplementary materials) to the originally proposed analysis.

However, the authors have also argued in their rebuttal letter that the inclusion of acoustic analysis was always a post-hoc decision and they note that any analysis is likely to be underpowered. This then speaks against conducting any acoustic analyses as underpowered analyses are unlikely to be informative.

Overall, I would encourage the authors to more clearly note their reservations about a lack of power to explain the null findings for the existing acoustic analyses. I'll leave the decision whether to request the inclusion of a more traditional acoustic analysis to strengthen their paper to the editor.

·

Basic reporting

The authors have done an excellent job revising the paper and addressing the comments and concerns of the Editor and all three reviewers, I applaud their effort and believe that the end product is a much more transparent and replicable piece of research. Below I enumerate my remaining concerns regarding stimulus pair creation and acoustic analysis.

Experimental design

I thank the authors for now uploading the questionnaire and all vocal stimuli as supplementary materials. However listening to these stimuli has raised a few additional concerns.

First, it was not evident from the Methods section or Table 1 that some of the screams are ‘re-used’ in more than one stimulus pair. For example, Stimulus pairs ‘Different Vocalizer 2’ and ‘Different Vocalizer 3’ contain the exact same first scream, followed by a different second scream. Listeners could have used this information (I’ve heard this scream before) when judging whether two screams in a pair are likely to have been produced by the same Vocalizer.

In the Methods and in Table 1, where currently the authors provide only ‘stimulus pair gender breakdown’, it should be made clear how many different ‘Vocalizers’ were included in the stimulus pairs and how many of the screams were re-used across pairs. This should be notes as a potential confound.

Second, listeners could have used vocal cues to age, and to a lesser extent scream context, to judge whether screams were produced by the same vocalizer. Vocalisers in ‘Different Vocalizer’ pairs were matched for “gender, relative age (child or adult), and apparent emotional context” (L 210). However the authors do not provide any additional information about the age ranges of vocalizers. In theory, “adult” could include ages 18 to 100. Even if the age difference between two vocalizer’s in a ‘Different Vocalizer’ pair was relatively smaller than this (e.g., 20 years), it is possible that listeners could still pick up on this age difference to inform their judgment. To my knowledge there is no evidence (yet) that listeners can judge rough age from screams, but it’s probable that they can, and this too should be flagged as a possible confound in the judgment task.

Similarly I worry that context (pain, fear, excitement) could have aided listeners in their judgment. While the authors note that screams in ‘Same Vocalizer’ stimulus pairs were either taken from “the same scream bout, or different screams from entirely separate bouts and eliciting contexts” (L 208), it seems likely that screams from the same individual would be taken from the same context more often than would screams from two different individuals.

Minor edits:

The authors indicated that, due to a coding error, two stimulus pairs were identical, i.e., contained the exact same first and second screams (L215). These pairs should be identified (and labelled) in the supplementary audio files.

Please indicate the degree of stretching (% from original) in the Duration Modified stimulus pairs (L 200).

In Table 1, the “Stimulus Type” category names still need to be revised (Same-A = Same Vocalizer, etc).

Validity of the findings

All three reviewers suggested that the authors provide additional descriptions and justifications for their acoustic analysis. As I noted in my original review, my biggest concern was the omission of key acoustic variables, namely F0, formants, and nonlinear phenomena (e.g., deterministic chaos) that are known to vary between individuals and to influence listeners’ voice-based judgments of a variety of speaker traits including identity, sex, age, body size, emotional and motivation states, etc. Formant frequencies are a key cue to identity in humans, as in other mammals, and F0 too appears to vary systematically between individuals, even in their nonverbal vocalisations (e.g., babies’ cries).

Omitting F0 and formants can be misleading. Based on a large amount of previous literature and strong theory-driven predictions, it is highly likely that these omitted acoustic parameters influenced (and thus would have predicted) listeners’ judgments, whereas the lesser-justified acoustic features that the authors chose to examine (bifurcations and harmonic regime) did not.

My recommendation is to either perform a comprehensive theory-driven acoustic analysis, or to remove acoustic analysis from the paper altogether. I appreciate that the authors’ main (and as they note, only) initial goal of the study was to test whether listeners can discriminate identity from screams. Their exploratory acoustic analysis was post-hoc. Perhaps this is all the more reason to remove the acoustic analysis from the paper altogether?

Identifying acoustic cues to identity in nonverbal vocalisations is clearly an important avenue for future work. Once the acoustic correlates of listeners’ judgments are identified, resynthesis techniques can be used to manipulate these acoustic parameters to test their direct, independent or interactive effects on listeners’ judgments and accuracy scores.

Reviewer 3 ·

Basic reporting

No comment.

Experimental design

No comment.

Validity of the findings

No comment.

Additional comments

The authors have substantially revised this manuscript in line with the suggestions from the reviewers and editor, resulting in a superior manuscript. In particular, I find that the additions and changes to the Introduction and Discussion have resulted in greatly improved narrative clarity, appropriately situating this work within the broader literature and providing a more thorough consideration of the outcomes of this analysis. The Analysis + Results sections are also considerably more transparent than in the previous version.

I am satisfied with the response to each of the comments provided in my original review and have no further suggestions.

---

## Round 0.3 · accepted · Accept

Thank you for your attention to these final few details. I am pleased to accept the manuscript for publication. I look forward to reading about the follow-up study. Do consider us for that manuscript when the time comes!